# Syllable processing is organized in discrete subregions of the human superior temporal gyrus

Daniel R. Cleary [1,2]*, Youngbin Tchoe[3], Andrew Bourhis[3], Charles W. Dickey[4], Brittany Stedelin[2], Mehran Ganji[3], Sang Heon Lee[3], Jihwan Lee[3], Dominic A. Siler[2], Erik C. Brown[2], Burke Q. Rosen[5], Erik Kaestner[6], Jimmy C. Yang[7,8], Daniel J. Soper[7], Seunggu Jude Han[2], Angelique C. Paulk[7,9], Sydney S. Cash[7,9], Ahmed M. Raslan[2☯], Shadi A. Dayeh[3,10,11☯], Eric Halgren[12,13☯]

1 Department of Neurosurgery, University of California San Diego, La Jolla, California, United States of America, 2 Department of Neurological Surgery, Oregon Health & Science University, Portland, Oregon, United States of America, 3 Department of Electrical and Computer Engineering, University of California San Diego, La Jolla, California, United States of America, 4 Neurosciences Graduate Program, University of California, San Diego, La Jolla, California, United States of America, 5 Department of Neuroscience, Washington University School of Medicine, St. Louis, Missouri, United States of America, 6 Center for Multimodal Imaging and Genetics, University of California San Diego, La Jolla, California, United States of America, 7 Department of Neurology, Massachusetts General Hospital, Boston, Massachusetts, United States of America, 8 Department of Neurosurgery, Massachusetts General Hospital, Boston, Massachusetts, United States of America, 9 Center for Neurotechnology and Neurorecovery, Department of Neurology, Massachusetts General Hospital, Harvard Medical School, Boston, MA, United States of America, 10 Materials Science and Engineering Program, University of California San Diego, La Jolla, California, United States of America, 11 Department of Bioengineering, University of California San Diego, La Jolla, California, United States of America, 12 Department of Radiology, University of California San Diego, La Jolla, California, United States of America, 13 Department of Neuroscience, University of California San Diego, La Jolla, California, United States of America

☯ These authors contributed equally to this work.

* clearyd@ohsu.edu

**Data Availability Statement:** Data are available: https://zenodo.org/records/12792343.

**Funding:** This work was supported by National Institutes of Health (NIH, https://www.nih.gov),

## Abstract

Modular organization at approximately 1 mm scale could be fundamental to cortical processing, but its presence in human association cortex is unknown. Using custom-built, high-density electrode arrays placed on the cortical surface of 7 patients undergoing awake craniotomy for tumor excision, we investigated receptive speech processing in the left (dominant) human posterior superior temporal gyrus. Responses to consonant-vowel syllables and noise-vocoded controls recorded with 1,024 channel micro-grids at 200 μm pitch demonstrated roughly circular domains approximately 1.7 mm in diameter, with sharp boundaries observed in 128 channel linear arrays at 50 μm pitch, possibly consistent with a columnar organization. Peak latencies to syllables in different modules were bimodally distributed centered at 252 and 386 ms. Adjacent modules were sharply delineated from each other by their distinct time courses and stimulus selectivity. We suggest that receptive language cortex may be organized in discrete processing modules.

National Science Foundation (NSF, https://www. nsf.gov), and NIH High-Risk, High-Reward Research program (https://commonfund.nih.gov/ highrisk), in the form of the following awards: NIH Brain Initiative, #MH120886-01, DRC; NIBIB award #DP2-EB029757, SAD; NIH BRAIN Initiative R01NS123655, SAD; NIH BRAIN Initiative UG3NS123723, SAD; NIH BRAIN Initiative R01DA050159, SAD; NSF award #1728497, SAD; NIMH R01 MH117155, EH; NIH award #K24-NS088568, SSC; MGH Executive Committee on Research, SSC; Tiny Blue Dot Foundation, SSC, ACP. The funders had no role in study design, data collection and analysis, decision to publish, or preparation of the manuscript.

**Competing interests:** SD and AR have equity in a startup company (Intelecterra). SC has a financial interest in an unrelated company (Beacon Biosignals). No other authors report competing interests.

**Abbreviations:** ERPAC, event-related phase amplitude coupling; ERSP, event-related spectral power; HG, high gamma; NNMF, non-negative matrix factorization; PLV, phase-locking value; SOA, stimulus onset asynchrony; STG, superior temporal gyrus.

Cortical function is often understood as composed of discrete processing modules whose outputs converge on other modules. Commonly, cortical columns are posited to perform this role, bridging local circuits and cortical parcels. In primary sensory cortices of certain species, spatially demarcated groups of neurons form discrete columns that respond to specific stimuli, distinct from neighboring columns [1–3]. Using fMRI, orientation-selective and ocular dominance columns have been identified in V1 [4], color and binocular disparity stripes in V2 and V3 [5], axis of motion columns in MT [6], and frequency columns in A1 [7]. Column diameter varies. They are approximately 500 μm in cat somatosensory [8] or macaque ocular dominance, but human ocular dominance are approximately 1.7× wider [9]. However, due to limited spatial resolution, these fMRI studies have difficulty demonstrating a mosaic organization with functional properties relatively constant within the column and changing sharply at their boundaries. Furthermore, with the partial exceptions of MT and V2/3, these findings have been restricted to primary sensory cortex. Although columnar organization is clear anatomically [10,11], it has not been associated with functional columns [12]. Thus, the status of columns, or other modularity at that scale, in human association cortex, remains unknown.

Direct intracranial recordings from the dominant posterior superior temporal gyrus (STG) have characterized the response correlates of neuronal populations to speech [13]. Generally, population firing is estimated from high gamma amplitude (HG; approximately 70 to 190 Hz), recorded from approximately 3 mm diameter contacts at 10 mm pitch. Using 4 mm pitch, Mesgarani and colleagues [14] found HG selectivity for groups of phonemes that shared articulatory features, and Leonard and colleagues [15] found that STG neurons in a track perpendicular to the surface often encode similar features, and their firing is reflected in the overlying surface. However, lateral spatial resolution in these studies has not been sufficient to resolve cortical columns.

Here, we measured the shape, size, and spacing of cortical modules selectively engaged by syllables using microgrids on the pial surface of STG. For the first set of experiments (4 participants), we used a 128-channel grid with 2 × 64 contacts at 50 μm pitch (Fig 1A). The second set of experiments (3 participants, one with 2 placements) used a 1,024-channel 16 × 64 grid at 200 μm pitch (Fig 1B). All participants were undergoing acute awake craniotomies for mapping eloquent cortex prior to removal of a tumor [16]. Arrays were placed under visual control on a relatively avascular cortical surface which previous clinical testing identified as eloquent (Fig 1C and 1D). Patients listened to consonant-vowel syllables and detected those that formed valid words (words and pseudo-words were combined for analysis). Noise-vocoding of the same stimuli produced controls with the same amplitude envelope in different frequency bands as the spoken stimuli (Fig 1E).

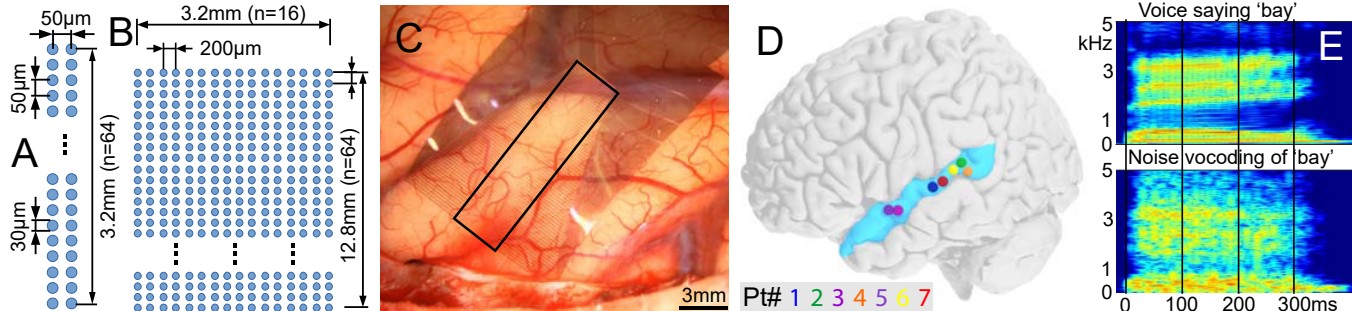

**Fig 1. Electrodes and stimuli used in the preliminary experiments.** Layouts of (**A**) 128 ch array (2 × 64, 50 μm pitch, 30 μm diameter PEDOT contacts) and (**B**) 1,024 ch array (16 × 64, 200 μm pitch, 30 μm diameter Pt nanorod contacts). (**C**) Intraoperative photo of 1,024 ch array adhering to the pial surface. (**D**) Approximate centers of electrode arrays on standard brain. (**E**) Spectrograms of an example word and its noise-vocoded control.

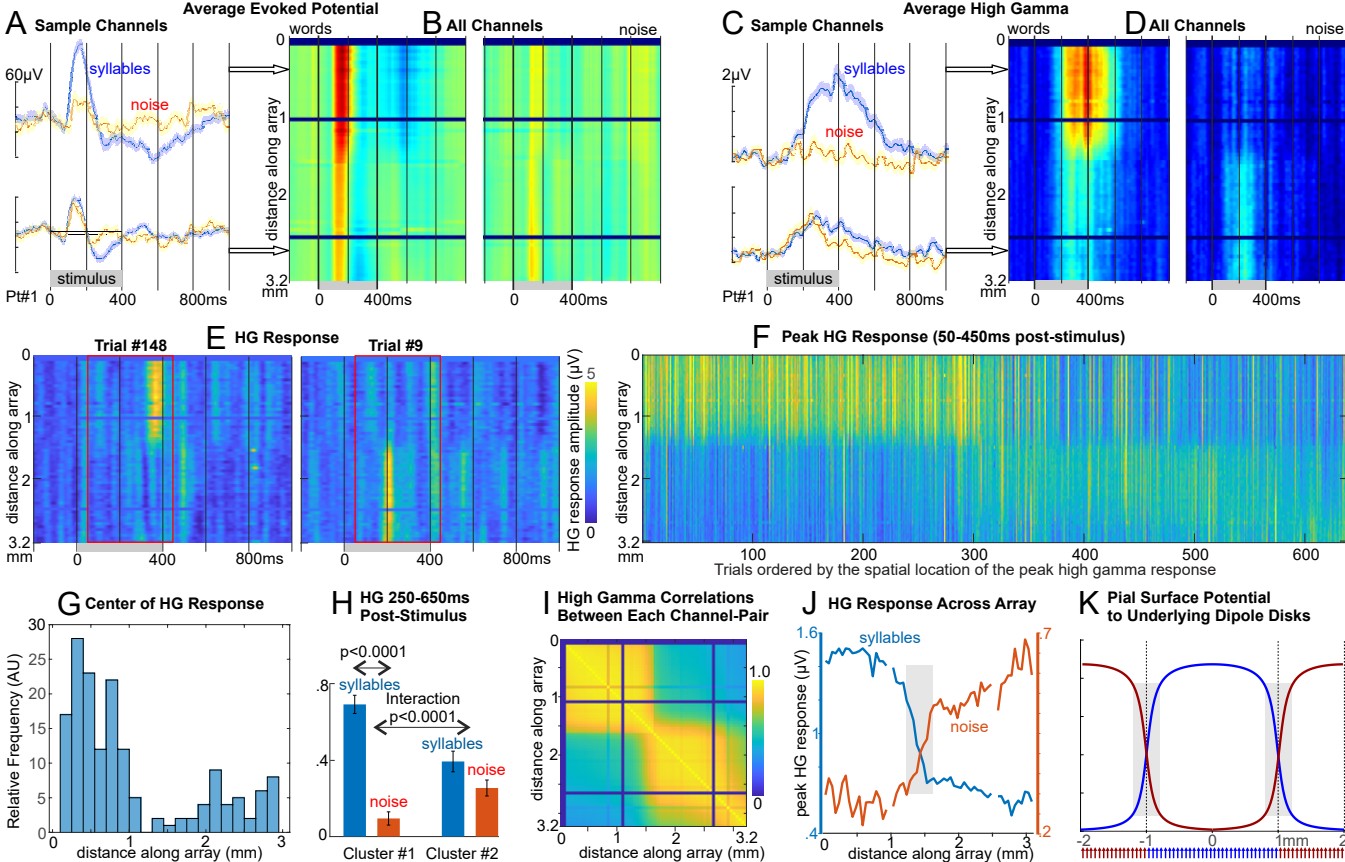

**Fig 2. Selective responses to syllables.** (**A**) Averaged evoked potential from 2 sample channels shows greater selectivity in upper channels, with (**B**) an abrupt change in response midway through the array. (**C, D**) Same as A, B but analytic amplitude high gamma (70–190 Hz) averaged across trials. Note that the upper channels have very consistent responses, as do the lower, suggesting 2 distinct functional modules. (**E**) As in C, but for 2 individual trials showing spatially disjoint HG burst-responses. (**F**) The peak responses from each trial are plotted in accordance by the spatial middle of the response. Note the highly consistent co-activation of each module across trials. (**G**) A histogram of the spatial distribution of gamma responses shows the bimodal distribution. (**H**) ANOVA finds a syllable-selective HG response in module #1 and an interaction of syllable with module (S1 Data). (**I**) HG-band activity in the first approximately 25 contacts (spanning approximately 1.25 mm) were highly correlated over the entire task with each other, then after a transition zone of approximately .4 mm (10%–90% levels), the next 25 contacts were again highly correlated with each other but not with the first 25. Thus, the channels were organized into 2 groups, with very high within-group, and low out-group, correlation. (**J**) Blue lines show the peak value of the HG average across all Syllable trials, at each contact; Red lines Noise trials. Note the roughly flat plateau in the modules and a sharp transition zone. (**K**) The surface potential produced by alternating 2 mm diameter discs of dipoles placed 100 μm below the surface. The modeled curves are similar to the empirical data in panel G, suggesting possible mosaic organization of functional domains in human association cortex. The gray bars are 400 μm wide and span the distance over which the curves pass from 10% to 90% of maximum, marking the transition zone. Note that the 3.2 mm of the pial surface spanned by this 2 × 64 array at 50 μ pitch would sample activity from 2 or 3 modules with approximately 1.7 mm diameter. Similar results from other patients are shown in S1–S4 Figs.

Six of the 7 patients showed significant modulation of local field potentials to syllables versus noise, in both lower frequencies (as the average evoked potential, Fig 2A) and as the average instantaneous high gamma amplitude (HG; Fig 2C). In one example with a linear 2 × 64 arrays, both the evoked potential and HG responses were very similar for approximately 25 channels (approximately 1.3 mm) and then abruptly transitioned to channels with distinct response properties which again were similar to each other (Fig 2B and 2D). For HG, this effect was clear not only in the average (Fig 2B) but in individual trials where the response appeared as a simultaneous burst of HG in consistent clusters of adjacent channels (Fig 2E–2G). Similarly, responding channels were clustered using non-negative matrix factorization (NNMF) of the gamma-band (70 to 190 Hz) analytic amplitude, and the responses quantified for each cluster of channels (2H). Within these cortical "modules," gamma-band correlations plateaued at a

very high level, but at the boundary between modules, correlations dropped rapidly (Fig 2I). The consistency of responses within modules and the steep transition between modules was also apparent in other measures, including the peak value of the HG to Syllables or Noise at each location (Fig 2J). These empirical curves show a transition zone between modules of 400 μm width, which corresponds to that which would be generated by a disc of dipoles 100 μm below the cortical surface (Fig 2K).

Co-active modules were also identified in the 1,024-channel arrays using NNMF of the gamma band (70 to 190 Hz) analytic amplitude recorded over the entire task period (Fig 3A and 3E). Since the algorithm has no access to spatial information, channels are spatially co-localized only because of functional similarity in signals. HG band phase-locking is very high and consistent within modules and drops rapidly at their boundaries (Fig 3C and 3D). The average diameter of modules with significant selectivity for syllables of approximately 1.7 mm was roughly consistent across the 3 patients, 4 recordings, and 27 modules (mean ± stdev height: 1.73 ± 0.64 mm; width: 1.73 ± 0.59 mm).

In some patients, all modules have very similar time courses, and thus their delineation is unlikely to reflect differences in temporal response patterns. In order to further evaluate the effect of temporal pattern, we repeated NNMF including only the time-averaged responses from 100 to 400 ms in each trial, instead of the entire recording. Despite the approximately 1,500-fold decrease in samples, similar modules were identified (S9 Fig). We conclude that while it is possible that time-course differences made some contribution to refining module boundaries, such differences are not generic auditory on-off/sustained-phasic response modulations but rather modulation within the CV syllable.

Compared to the 100 ms pre-stimulus baseline, HG from 50 to 450 ms in all 27 modules was significantly greater to the stimuli ($p < 0.02$, two-tailed paired $t$ test, FDR-corrected). While 18 of 27 modules responded preferentially to syllables, 1 responded more to noise, and 8 did not significantly distinguish syllables and noise ($p < 0.02$, two-tailed paired $t$ test, FDR-corrected). Examination of contacts at module boundaries revealed a sharp transition between adjacent contacts separated by 200 μm (Fig 3B and 3C). Peak response times in the modules were bimodal, with 13 peaking below 350 ms and 14 above (Fig 3F–3H). The histogram of peak latencies was found to be marginally more parsimoniously modeled by a mixture of 2

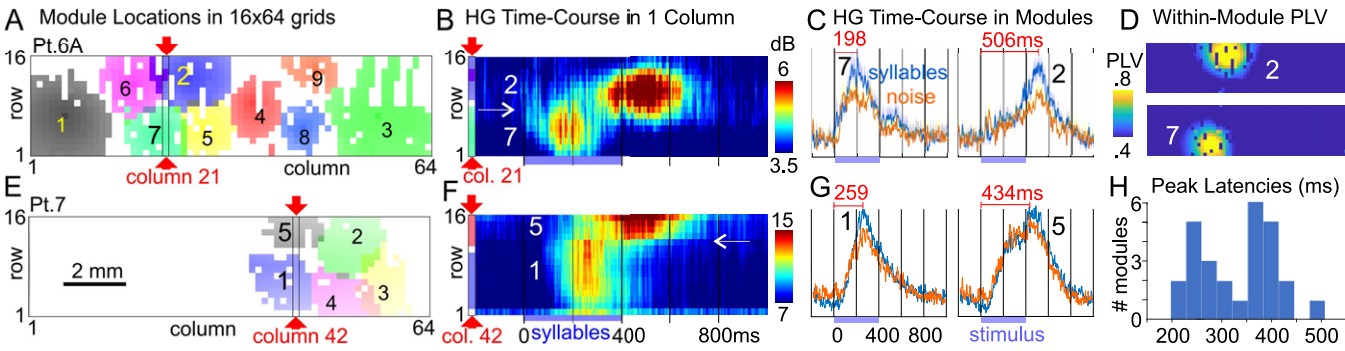

**Fig 3. Discrete functional modules evoked by CV syllables on STG.** (**A**) Modules of co-active electrodes at 200 μm pitch in 16 × 64 arrays identified with NNMF of HG amplitude recorded over the entire task period, in the first placement in patient 6. Module boundaries are at 2 stdev above the mean of the entire array. (**B**) HG amplitude recorded from column 21 of placement 6A, passing through module 7 which peaks at approximately 200 ms after stimulus onset, and module 2 which peaks at approximately 500 ms. Columns are marked with red arrows in panel A. Note that adjacent contacts belonging to different modules have completely different time courses of activation (→). (**C**) Average HG amplitude to syllables and noise for modules 2 and 7. (**D**) PLV [19] between HG at the center contact of each module and the entire array. (**E–G**) Same as A–C but for a traverse in column 42 of modules 1 and 5 in patient 7. Again, these modules have different latency peaks (approximately 250 and approximately 450 ms), with a sharp transition between response profiles (←). (**H**) Histogram of peak module response latencies shows 2 distinct peaks. Similar results from other patients are shown in S5–S7 Figs.

gaussians than by one (Akaike information criterion = 314.08 versus 314.93) with means of 252 and 386 ms. The first phoneme of each stimulus was a consonant and the second a vowel, and the latency between the average onsets of the consonants and vowels were similar to those of the early and late module response peaks. Additional recordings will be needed to test the hypothesis that consonants and vowels are encoded in adjacent sharply delineated modules.

Based on these findings, we can estimate that each typical 3 to 4 mm diameter clinical electrode would record from 4 to 8 modules, and at 10 mm spacing would miss approximately 85% of the modules. Conversely, approximately 250 μm diameter contacts at approximately 1 mm pitch would result in recordings from all modules, in most cases without significant crosstalk. Although these estimates are very rough, and assume that the modular organization exists in other association areas and are the same size, they suggest that finer sampling may reveal important clinical [17] and basic information, and provide the approximate spatial sampling necessary in language prostheses for detecting all functional units.

In conclusion, high-resolution recordings from STG provide evidence for modular organization of functional units having: (1) roughly circular shapes with approximately 1.7 mm diameter; (2) sharp edges with immediately adjacent modules having contrasting response time courses and stimulus correlates; (3) roughly uniform size; and (4) continuous mapping over the sampled cortical surface forming a mosaic. Although these characteristics are consistent with a columnar organization, convincing demonstration would require oblique laminar recordings identifying a sharp transition at depth. Modular organization has also been seen for functional units with fMRI, for example for faces, but those are typically irregular approximately 5 to 20 mm patches, and do not "pave" the cortex with multiple modules with sharp transitions [18]. It is possible to tile the entire cortical surface with approximately 128,000 1.7 mm diameter modules, each with approximately 96,000 neurons. Investigation of whether such modules exist in other association cortices, their dimensions, and information-processing activities of modules may inform multiresolution models of cortical function.

## Supplementary information

### Online methods

*Acute intraoperative setting.* When a tumor or lesion is near eloquent areas, neurosurgeons may elect to perform a resection procedure while the patient is awake, primarily using local anesthetic and systemic analgesics while minimizing sedation [16]. This technique allows the surgeon to closely monitor brain function during surgery, ensuring maximum safe resection while avoiding damage to important areas. This provides a unique opportunity for research on functional cortex while the patient is awake and interactive. Awake craniotomies, however, are relatively infrequent surgeries, and the potential time for research is limited to 20 min. Patients gave written informed consent to the microgrid recordings under a protocol approved by the local Institutional Review Board (OHSU IRB # 19099) and in accordance with the principles expressed in the Declaration of Helsinki.

*Microarrays* [20,21] have ultra-low impedance 30 μm diameter PEDOT:PSS (typical 1 kHz impedance magnitude of 38 kΩ) and Pt nanorod contacts (typical 1 kHz impedance magnitude of 25.5 kΩ) on a 7-μm thick parylene C substrate that conforms to the pial surface, further facilitating high-quality recordings. The fabrication and testing of these microarrays have previously been reported [20–22]. All devices in the experimental portion are powered via battery to avoid risk of ground loops or mains flow to the patient [23].

*LFP preprocessing* is performed in MATLAB 2019b and LFPs inspected visually using the FieldTrip toolbox [24], and 60 Hz and harmonics are notch filtered (zero-phase).

*Time-frequency analyses*. Average time-frequency plots of the event-related spectral power (ERSP) are generated from the broadband LFP using EEGLAB [25]. ERSP is calculated from 1 Hz to 500 Hz with 1 Hz resolution by computing and averaging fast Fourier transforms with Hanning window tapering. Each 1 Hz bin is normalized with respect to the mean power at −200 to −150 ms and masked with two-tailed bootstrapped significance with FDR correction and $\alpha$ = 0.05. Event-related phase amplitude coupling (ERPAC) is calculated using an open-source MATLAB package [26].

*Non-negative matrix factorization (NNMF)* of high gamma (70 to 190 Hz) analytic amplitude recorded over the entire task period at 1 ms accuracy was used to cluster the channels based on signal similarity without regard for spatial colocalization of channels [27]. The entire unaveraged signal from all technically correct channels over the entire task was entered into the factorization. The optimal number of factors for NNMF was selected by picking the approximate nadir using cross-validation; one by one, a single element is left out of the data set, and the prediction error is measured for each possible number of modules [28]. Once the optimal number of modules is defined, for each module the algorithm creates a map by assigning weights to each channel. Modules are defined by spatial colocalization of highly weighted channels; the threshold for channel inclusion is defined as weights 3 standard deviations above the mean for all channels for that module. Colocalized modules of channels are subsequently refined using phase-locking value, wherein the spatial center of each module was identified, and phase-locking calculated for each channel relative to the center channel. To display the boundaries of modules, a threshold value of PLV is picked that minimizes the overlap between modules.

The NNMF analysis was repeated including only the time-averaged responses from 100 to 400 ms in each trial, instead of the entire recording at 1 ms in order. The time-averaged NNMF used only approximately 200 samples for each of the approximately 1,000 channels (one for each trial), whereas the original method entered approximately 300 k samples per channel (approximately 5 min at 1,000 Hz). Inasmuch as factorial analyses of this kind are well known to require a large number of samples, it is likely that the differences between the panels are due to both the decreased data entering the time-averaged NNMF as well as the specific absence of temporal information.

We explicitly tested if the modules identified by NNMF could be due to cross-talk in the ribbon cable running from the electrode to the electronics, or in shared processing characteristics in the electronics (S8 Fig). First, we plotted a histogram of the physical distances between contacts belonging to the same module (S8A Fig). As expected, given that modules are compact spatially, these formed a roughly Gaussian distribution centered at approximately 1 mm and never exceeding approximately 2.3 mm. In contrast, the histogram of lines between contacts belonging to the same module had 2 main peaks, one at short distances mainly approximately 1 to 8 separations, and one at approximately 800 to 1,000 separations (S8B Fig). A close-up of the short distances (S8C Fig) shows a distribution completely different from S8A Fig. If the modules were due to crosstalk then the corresponding long and short distances should have been noted in S8A Fig, but they were not. Similarly, the assignment of channels to banks of electronics generally alternated between rows of contacts (S8D Fig), so if shared electronic processing characteristics caused the module clustering, individual clusters should then also have members on alternate rows, which they did not. Recall that the NNMF procedure is completely unaware of the spatial location of the contacts, thus if cross-talk in the ribbon cable, or shared amplifier bank characteristics underlay the modules, then they would have reproduced the proximity matrices of the cable or of the amplifier assignments, rather than being organized in spatially compact roughly circular domains. Finally, we also note that the same ribbon-cable and amplifier bank assignments were used in all experiments and yet quite

different module locations were observed. Furthermore, in one patient, the same actual array was simply moved from one STG site to another between blocks of the task, resulting in different module locations being observed on the same array. This evidence is inconsistent with modules arising from technical sources of signal similarity.

*Phase-locking value*. PLV is an instantaneous measure of phase-locking [19], which unlike coherence is not affected by shared amplitude modulation. PLV time courses are computed using the analytic angle of the Hilbert transformed 70 to 190 Hz bandpassed (zero-phase shift) signals of each channel pair, and their significance calculated using FDR correction and a null distribution [29].

*Statistical analyses*. HG relative to behavioral events is compared to an appropriate baseline control period, and between task conditions. To determine if a module was responsive to the stimuli, the Hilbert analytic amplitude from 70 to 190 Hz was averaged on each trial in a between baseline period (100 ms before stimulus onset) and in the active period (50 to 450 ms poststimulus onset). A two-tailed paired $t$ test was performed using pairs of values from each trial. To determine if a module was differentially responsive to syllables versus noise, a two-tailed $t$ test was performed comparing the average value in the active period between syllable and noise trials. All statistical tests are evaluated with $\alpha = 0.02$, FDR-corrected across all patients and modules [29]. Patient 7 showed responses to both syllables and noise but they were not significantly different, possibly due to residual anesthesia that was applied during the preceding craniotomy, which was observed behaviorally as slow responses and intermittent confusion. Patient 7's five modules failed to distinguish between syllables and noise.

*Model*. The surface potential produced by alternating 2 mm diameter discs of dipoles (5,017 at 25 µ pitch, each 10e-6 µAmm) placed 100 µm below the surface in an infinite homogeneous isotropic medium (0.33 S/m). Estimates of number of modules and neurons per module based on total cortical surface area from ref [30] and total cortical neurons in gray matter from ref [31].

*Task*. A wide range of Consonant-Vowel syllables were presented to provide diverse activation of STG. Eight consonants were drawn equally from 4 categories following linguistic categories: Obstruent-Plosive (d, g), Obstruent-Fricative (s, z), Sonorant-Liquid (l, r), and Sonorant-Nasal (m, n). Nine vowels are drawn equally from 3 categories: Type 1 (u, aɪ, ɔ), Type 2 (ə, I, ɛ), and Type 3 (eɪ, æ, aʊ). Each possible CV combination is equally represented in the total stimulus set, resulting in 72 total syllables. Each was spoken once by a male voice and once by a female. Each presented sound lasts approximately 450 ms, and the SOA (stimulus onset asynchrony) was individually adjusted but usually approximately 1,100 ms. Stimuli were presented using Presentation (Neurobehavioral Systems) in approximately 6 min blocks of 288 trials, each stimulus occurring once. The task was to detect CV syllables which are syllables.

*Noise-vocoding*. For every CV syllable, a matching noise-vocoded stimulus was constructed taking the existing biphoneme stimuli and creating a 6-band stimulus in which white noise was multiplied by power in each of the bands to create a matched set of auditory stimuli with identical time-varying spectral acoustics [32,33]. Noise-vocoded stimuli preserve temporal envelope cues in each spectral band, providing a control for the sensory characteristics of speech, but the spectral detail within each band was degraded.

## Supporting information

**S1 Fig. S1 through S4 Figs show recordings from the 2 × 64 channel linear array, at 50 µm pitch.** Data from only one of the 2 channels at each location in the long dimension are presented for clarity; the opposite channel was highly similar in all cases where both were technically sound. Solid, dark blue pixels indicate channels without usable signal. S1 Fig shows

spectral and spatiotemporal responses for the linear array in patient #4. (A) The mean event-related spectrogram of all trials shows a change in both the high gamma-band and low frequency after the onset of stimuli. (B) Event-related phase-amplitude coupling (ERPAC) [26] between 2–8 Hz and 70–100 Hz shows a stimulus-related link in response for all channels, as measured with the z-score for the ERPAC (top). FDR-corrected *p*-values (bottom) shows that the time points of statistical significance for the ERPAC are in the first few hundred milliseconds after the onset of the auditory stimuli. (C) A matrix of phase-locking values (PLV) between all channels shows 2 distinct regions of high phase-locking within but not between regions. (D) On the spatiotemporal response map, the entire electrode consistently responds to auditory stimuli, but 2 distinct clusters of channels are identified that factorize together (green bar, yellow bar). Close inspection shows a late temporal response in cluster #1 that separates it from cluster #2 (arrowhead). The same channels that cluster together in (D) show a high-degree of phase-locking in the gamma band (C). (E) The mean gamma-band response in the 50–450 ms after onset of stimuli for the clustered channels from (D) shows a stronger response to words than noise stimuli, which is only significant with cluster #2 (S2 Data). (EPS)

**S2 Fig. Spectral and spatiotemporal responses for linear array in patient #8, recording from non-dominant pSTG.** (A) The mean event-related spectrogram of all trials shows an event-related change in both the high gamma-band and low frequency after the onset of stimuli. (B) On the spatiotemporal response map of gamma activity, increased activity was observed in the few hundred milliseconds after onset of auditory stimuli at time zero. Two distinct spatial segments are identified using factorization to cluster the channels (green bar, yellow bar). (C) Quantifying the mean gamma-band response in the 50–450 ms after onset of stimuli for the clustered channels shows a stronger response to noise than word stimuli for both clusters of channels (S3 Data). (EPS)

**S3 Fig. Spectral and spatiotemporal responses for linear array in patient #3.** (A) The mean event-related spectrogram of all trials shows a lower amplitude of increase in both the high gamma-band and low frequency than observed in other patients. (B) Similarly, on the spatiotemporal response map of gamma activity, increased activity was only observed in one segment for in the few hundred milliseconds after onset of auditory stimuli. Only one spatial segment was identified using factorization (green bar). (C) Quantifying the mean gamma-band response in the 50–450 ms after onset of stimuli for the clustered channels shows a stronger response to words than noise stimuli (S4 Data). (EPS)

**S4 Fig. Spectral and spatiotemporal responses for linear array in patient #2.** (A) The mean event-related spectrogram of all trials shows a change in both the high gamma-band and in low frequencies after the onset of stimuli. (B,C) On the spatiotemporal response map of gamma activity for words, spatially segregated increase activity was observed in the few hundred milliseconds after onset of auditory stimuli at time zero. Three distinct spatial segments are identified using factorization to cluster the channels, two of which appear to preferentially respond to word stimuli (green bar, brown bar) while one segment appears to preferentially respond to noise stimuli (yellow bar). (D) Quantifying the mean gamma-band response in the 50–450 ms after onset of stimuli for the clustered channels shows a stronger response to words than noise stimuli for one of the clusters of channels (green cluster), a stronger response to noise for one cluster (yellow cluster), and an indeterminate response for one cluster (brown

cluster) (S5 Data).
(EPS)

**S5 Fig. Spatiotemporal maps of gamma-band responses for Patient #6A, using the 1,024-channel electrode array (16 × 64 with 200 μm spacing).** (A) The mean gamma-band response between 100 ms and 500 ms after the onset shows multiple poorly demarcated regions increased signal. (B) Finer temporal discrimination is used to view gamma-band responses every 100 ms, which shows distinct spatial regions of activation. The spatial regions have distinct temporal onset and offset. Arrowheads indicate spatial regions with unique temporal activation times. (C) A spatial map of the 16 × 64 channel electrode with the results of factorization overlaid. Factorization identified 9 distinct, spatially adjacent regions. (D) For an individual cluster of channels (Module #1), the phase-locking value for each cluster shows a high degree of concordance for spatially adjacent channels with rapid drop off. (E) The mean signal of the principal components of Module #1 shows the temporal and feature preference for the underlying channels. The mean response is calculated using the average response relative to the start of the stimuli for all trials. (F) The other modules identified by factorization similarly show distinct temporal properties in terms of average onset, offset, and response features.
(EPS)

**S6 Fig. Spatiotemporal maps of gamma-band responses for Patient #6B, using the 1,024-channel electrode array (16 × 64 with 200 μm spacing).** (A) A spatial map of the 16 × 64 channel electrode with the results of factorization overlaid. Factorization identified 9 distinct, spatially adjacent regions. (B) For an individual cluster of channels (Module #1), the phase-locking value relative for cluster shows a high degree of concordance for spatially adjacent channels with rapid drop off. (C) The mean signal of the principal components of Module #1 shows the temporal and feature preference for the underlying channels. The mean response is calculated using the average response relative to the start of the stimuli for all trials. (D) The other modules identified by factorization similarly show distinct temporal properties in terms of average onset, offset, and response features.
(EPS)

**S7 Fig. Spatiotemporal maps of gamma-band responses for Patient #7, using the 1,024-channel electrode array (16 × 64 with 200 μm spacing).** (A) A spatial map of the 16 × 64 channel electrode with the results of factorization overlaid. In this patient, factorization identified 5 distinct, spatially adjacent regions. (B) For an individual cluster of channels (Module #1), the phase-locking value relative for cluster shows a high degree of concordance for spatially adjacent channels with rapid drop off. (C) The mean signal of the principal components of Module #1 shows the temporal and feature preference for the underlying channels. The mean response is calculated using the average response relative to the start of the stimuli for all trials. (D) The other modules identified by factorization similarly show distinct temporal properties in terms of average onset, offset, and response features.
(EPS)

**S8 Fig. Comparison of spatial organization of modules to that expected if they were to arise from cross-talk in the ribbon-cable or shared processing electronics.** We explicitly tested if the modules identified by NNMF could be due to cross-talk in the ribbon cable running from the electrode to the electronics, or in shared processing characteristics in the electronics. (A) Histogram of the physical distances between contacts belonging to the same module. Note that, as expected given that modules are compact spatially, these formed a roughly Gaussian distribution centered at approximately 1 mm and never exceeding

approximately 2.3 mm. (B) Histogram of the number of conducting wires on the ribbon cable between contacts belonging to the same module. Note that this histogram has 2 main peaks, one at short distances with mainly approximately 1–8 wires between adjacent channels and another peak approximately 800–1,000 wires between adjacent channels. (C) Close-up of panel B shows a distribution clearly different than that observed for intra-module distances between contacts in Panel A. (D) Assignment of channels to banks of electronics (Analog-to-Digital Converter [ADC] and amplifiers). Note the striped/mosaic organization of the ADC, again clearly different from that of identified modules.
(EPS)

**S9 Fig. Non-negative matrix factorization weights derived from using only the mean of the poststimulus response, from patients 6A and 6B (also shown in Fig 3 and S5 Fig, respectively).** For each auditory stimulus, the analytic amplitude of the gamma-band response for each channel was averaged from 100 ms to 400 ms poststimulus, and this limited data set was fed into the NNMF algorithm. The algorithm receives no information on spatial locations of channels, but still reconstructs a similar spatial pattern of clustered channels as when the complete recording is used. (A) Example result showing a single cluster of spatially adjacent channels with high factorization weights from patient 6A when only using the average of the poststimulus gamma-band response. The lower number is the corresponding cluster number from Fig 3. (B) The other 8 clusters identified by NNMF based on the mean poststimulus response only, with corresponding cluster number from Fig 3 at bottom. (C) The clusters from A and B overlaid onto a single graph without overlap allowed between clusters. (D) For comparison, the clusters identified by using the entire recording at 1 ms resolution (previously shown in Fig 3). (E) As above, an example cluster of channels based on factorization weights using only the average of the poststimulus gamma-band response, this time from patient 6B. (F) The 8 other clusters identified by NNMF for the average response from patient 6B. (G) The clusters from E and F overlaid onto a single graph without overlap allowed between clusters. (H) For comparison, the clusters identified by using the entire recording at 1 ms resolution (previously shown in S9 Fig).
(EPS)

**S1 Data. The raw values for bar graphs shown in Fig 2H.**
(XLSX)

**S2 Data. The raw values for bar graphs shown in S1E Fig.**
(XLSX)

**S3 Data. The raw values for bar graphs shown in S2C Fig.**
(XLSX)

**S4 Data. The raw values for bar graphs shown in S3C Fig.**
(XLSX)

**S5 Data. The raw values for bar graphs shown in S4D Fig.**
(XLSX)

## Author Contributions

**Conceptualization:** Daniel R. Cleary, Youngbin Tchoe, Charles W. Dickey, Erik C. Brown, Angelique C. Paulk, Sydney S. Cash, Ahmed M. Raslan, Shadi A. Dayeh, Eric Halgren.

**Data curation:** Daniel R. Cleary, Youngbin Tchoe, Andrew Bourhis, Charles W. Dickey, Brittany Stedelin, Sang Heon Lee, Jihwan Lee, Dominic A. Siler, Erik C. Brown, Erik Kaestner, Jimmy C. Yang, Angelique C. Paulk, Ahmed M. Raslan, Shadi A. Dayeh, Eric Halgren.

**Formal analysis:** Daniel R. Cleary, Youngbin Tchoe, Andrew Bourhis, Charles W. Dickey, Burke Q. Rosen, Eric Halgren.

**Funding acquisition:** Daniel R. Cleary, Shadi A. Dayeh, Eric Halgren.

**Investigation:** Daniel R. Cleary, Youngbin Tchoe, Andrew Bourhis, Charles W. Dickey, Brittany Stedelin, Mehran Ganji, Sang Heon Lee, Jihwan Lee, Dominic A. Siler, Erik C. Brown, Jimmy C. Yang, Daniel J. Soper, Seunggu Jude Han, Angelique C. Paulk, Ahmed M. Raslan, Eric Halgren.

**Methodology:** Daniel R. Cleary, Youngbin Tchoe, Andrew Bourhis, Charles W. Dickey, Brittany Stedelin, Mehran Ganji, Sang Heon Lee, Jihwan Lee, Dominic A. Siler, Erik C. Brown, Burke Q. Rosen, Erik Kaestner, Jimmy C. Yang, Daniel J. Soper, Seunggu Jude Han, Angelique C. Paulk, Sydney S. Cash, Ahmed M. Raslan, Eric Halgren.

**Project administration:** Daniel R. Cleary, Sydney S. Cash, Shadi A. Dayeh, Eric Halgren.

**Resources:** Sang Heon Lee, Sydney S. Cash.

**Supervision:** Sydney S. Cash, Ahmed M. Raslan, Shadi A. Dayeh, Eric Halgren.

**Writing – original draft:** Daniel R. Cleary.

**Writing – review & editing:** Daniel R. Cleary, Eric Halgren.

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
