## [Editor Report · Decision Letter 0]

8 Feb 2024

Dear Dr Cleary, 

Thank you for submitting your manuscript entitled "Modular Phoneme Processing in Human Superior Temporal Gyrus" for consideration as a Short Reports by PLOS Biology.

Your manuscript has now been evaluated by the PLOS Biology editorial staff as well as by an academic editor with relevant expertise and I am writing to let you know that we would like to send your submission out for external peer review.

Once your full submission is complete, your paper will undergo a series of checks in preparation for peer review. After your manuscript has passed the checks it will be sent out for review. To provide the metadata for your submission, please Login to Editorial Manager (https://www.editorialmanager.com/pbiology) within two working days, i.e. by Feb 10 2024 11:59PM.

Kind regards,

Christian

Christian Schnell, PhD

Senior Editor

PLOS Biology

cschnell@plos.org

---

## [Decision Letter · Decision Letter 1]

4 Apr 2024

Dear Dr Cleary,

Thank you for your patience while your manuscript "Modular Phoneme Processing in Human Superior Temporal Gyrus" was peer-reviewed at PLOS Biology. It has now been evaluated by the PLOS Biology editors, an Academic Editor with relevant expertise, and by several independent reviewers. 

In light of the reviews, which you will find at the end of this email, we would like to invite you to revise the work to thoroughly address the reviewers' reports.

As you will see below, the reviewers think that the study is overall well executed and provides important insights. However, both reviewers raise concerns about missing details and the phoneme-selectivity aspect that need to be addressed. 

Given the extent of revision needed, we cannot make a decision about publication until we have seen the revised manuscript and your response to the reviewers' comments. Your revised manuscript is likely to be sent for further evaluation by all or a subset of the reviewers.

**IMPORTANT - SUBMITTING YOUR REVISION**

*Re-submission Checklist*

*Published Peer Review*

*PLOS Data Policy*

*Blot and Gel Data Policy*

Sincerely,

Christian

Christian Schnell, PhD

Senior Editor

PLOS Biology

cschnell@plos.org

REVIEWS:

Reviewer #1: SUMMARY

The authors used microarrays with very fine-grained spatial resolution. They recorded neural responses in superior temporal gyrus from 7 patients while they listened to consonant-vowel syllables in clear and noisy (via noise vocoding) conditions. They analyzed the similarity of responses from electrodes at variable distances in order to test for and describe the modular structure that emerges. They found that responses within ~1.7mm "modules" were very similar, which transition to a different response in the neighboring modules. This was replicated when using a number of different analysis methods, including supervised and non-supervised approaches. The authors conclude that the similarity structure they observe provides evidence for discrete processing modules along STG, with a consistent size, shape and density across patient recordings. 

Overall, this is important research that I think will help propel human neuroscience into a new era of spatially resolved recordings and important discoveries.

I noted some weaknesses that need to be addressed before I can fully endorse this article for publication. 

MAJOR 

[1] A big claim of the current paper is in identifying the size and shape of "processing modules". I would like to see some sanity checks that the design of the micro-array itself would not be biased to finding auto-correlation structure of that reported in the study. This could be achieved by analyzing the auto-correlation structure in the noise of the micro-arrays, and conducting dummy-recordings on agar or similar. 

[2] Another claim here is that the authors find "phoneme-selectivity". However, the authors conduct no analyses that support this claim directly. Phase-locking is indirectly related to the fact that the consonants occurred before the vowels across all stimuli. But a delayed response to a consonant and an immediate response to a vowel could result in the same phase-locking value. If the authors would like to make a claim about phonetic encoding, I suggest that they provide an analysis of the responses based on the factorial design of their stimuli {plosive, fricative, liquid, nasal} consonants and {low, medium, high} vowels.

[3] The patients hear CV syllables, which will sometimes form valid English words (e.g., "seɪ") and sometimes will not (e.g., "zeɪ"). The task of the participants is to press a button when they recognize a word of English, but the authors analyze responses to all CV syllables, not just those that are valid words. I ask the authors to clarify in the manuscript that analyses were performed on clear syllables and noise-vocoded syllables, where some items were words and some were not. Furthermore, to change the corresponding labels on the figures, and change them from "words" to "syllables" — or, similar. 

[4] There were a few sentences in the abstract that I found unclear. I request that the following are re-phrased to be clearer what is meant:

"modular organization fundamental to cortical processing" — at what spatial scale are you referring to here? 

"its presence in human association cortex is unknown" — if modular organization is critical, then how can its presence be unknown? Again, what spatial scale is being referred to? 

"characterized phoneme processing" — relating to point (2) above, I don't agree that this is an accurate summary of what was characterized in this study 

"phoneme selectivity" — see point (2) above 

MINOR

In Figure 1C, it is not clear to me what is the array, and what is the shadow of another object. 

In Figure 1D, why are single electrode positions visualized, when the authors use an entire array? 

In Figure 3E and elsewhere, why are there "empty" regions where no modules emerge? 

Reviewer #2 (Yves Boubenec): In this work, the authors tested the putative existence of functional modules within human auditory cortex. They show that functional responses are shared within regions of a couple of mm in diameter, suggesting discrete processing modules. That is a fascinating results that echo with long-standing literature of columnar processing in visual cortex.

The study is concise and compact, which is pleasant, but makes it missing a few points for clarity. Little, if none, is said about phonetic/syllabic tuning for these cortical domains.

major comments:

From figure 3, it looks like the factorization algorithm grouped channels mainly by the temporal profile of the response time-courses. This is indeed an important feature of functional responses. Nevertheless, these columns show strong speech responses, and one should not ignore the tuning of these channels.

1) Does the clustering depend on using the time-course? Or are similar results found when using the average response to each sound (ignoring the time-course)?

2) As a matter of fact, I could not find much detail in the methods about the actual data provided to the factorization algorithm. Does it simply perform on the average response time-course? On the average time-course for noise and syllable stimuli? On the average time-course for each syllable? Or on the single-trial responses?

3) A clear link with previous literature is missing. What is the phonetic tuning of these modules? Is it consistent with results of Mesgarani's and Chang's teams (cited in the introduction)? Do they reveal another level of organization that previous studies could have missed (like each module being heavily dedicated to one specific group of phonemes)?

An alternative option is that, these modules could be poorly tuned to phonemes themselves, but instead exhibit syllabic tuning. And the data of this study can address this issue.

minor comments:

i) I was unclear about the exact stimulus set: do you confirm there are 24 distinct syllables in total?

ii) figures 2 IJK are not referenced. Reference to figure 2H refers in fact to figure 2K.

iii) my understanding is that no actual word was played. Instead, stimuli were single syllables. I suggest replacing 'words' in the figures by 'syllables'.

iv) acronym PLV (for phase locking) in the caption comes out of the blue.

---

## [Decision Letter · Decision Letter 2]

26 May 2024

Dear Dr Cleary,

Thank you for your patience while we considered your revised manuscript "Modular Phoneme Processing in Human Superior Temporal Gyrus" for consideration as a Short Reports at PLOS Biology. Your revised study has now been evaluated by the PLOS Biology editors, the Academic Editor and the original reviewers. 

In light of the reviews, which you will find at the end of this email, we are pleased to offer you the opportunity to address the remaining points from the reviewers in a revision that we anticipate should not take you very long. Please either conduct the suggested analysis or provide compelling argumentation why the existing analyses obviate the need for that.

We will then assess your revised manuscript and your response to the reviewers' comments with our Academic Editor aiming to avoid further rounds of peer-review, although might need to consult with the reviewers, depending on the nature of the revisions.

**IMPORTANT - SUBMITTING YOUR REVISION**

*Resubmission Checklist*

*Published Peer Review*

*PLOS Data Policy*

*Blot and Gel Data Policy*

Sincerely,

Christian

Christian Schnell, PhD

Senior Editor

PLOS Biology

cschnell@plos.org

REVIEWS:

Reviewer #1: The authors have addressed my concerns - thank you. 

Reviewer #2 (Yves Boubenec): I thank the authors you for their response and the revisions made to the manuscript. However, I remain concerned that my main critique has not been fully addressed. Specifically, there is still no analysis demonstrating that the NNMF leverages differences in responses across phonemes rather than primarily grouping channels based on generic temporal response patterns (e.g., onset or sustained responses).

To adequately address this point, I propose redoing the analysis using the time-averaged response (ignoring the time-course) for each sound. This would help determine whether the clustering observed is truly due to differences in phoneme-specific responses or if it is primarily driven by the temporal characteristics of the responses. Demonstrating that similar modules are identified using time-averaged responses would strengthen the argument that the NNMF is capturing phoneme-specific tuning rather than generic temporal response patterns.

Without this analysis, it remains unclear whether the observed clustering reflects phoneme-specific processing or if it is an artifact of the temporal dynamics of the neural responses. Therefore, I strongly recommend conducting this additional analysis to address this critical aspect of the study.

---

## [Decision Letter · Decision Letter 3]

12 Jul 2024

Dear Dr Cleary,

Thank you for your patience while we considered your revised manuscript "Modular Phoneme Processing in Human Superior Temporal Gyrus" for publication as a Short Reports at PLOS Biology. This revised version of your manuscript has been evaluated by the PLOS Biology editors, the Academic Editor and one of the original reviewers.

Based on the reviews , we are likely to accept this manuscript for publication, provided you satisfactorily address the following data and other policy-related requests.

* We would like to suggest a different title to improve the accessibility for our broad audience: "Syllable processing is organized in discrete subregions of the human superior temporal gyrus"

* Please add the links to the funding agencies in the Financial Disclosure statement in the manuscript details

* "SD and AR have equity in a startup company (Intelecterra)." is listed twice in the Competing Interests section. Please also state whether the other co-authors have any conflicts of interest.

* In the Methods section, please provide information on whether consent was given in written or oral form and whether the study was conducted according to the principles expressed in the Declaration of Helsinki.

* The abstract is very short and not very informative. Please provide a more comprehensive abstract that gives more context and specific information of what was done and its implications, as it is the only thing most readers will access. 

* DATA POLICY:

Regardless of the method selected, please ensure that you provide the individual numerical values that underlie the summary data displayed in the following figure panels as they are essential for readers to assess your analysis and to reproduce it: 2H, S1E, S2C, S3C and S4D.

* CODE POLICY

We expect to receive your revised manuscript within two weeks. 

*Published Peer Review History*

*Press*

Sincerely,

Christian

Christian Schnell, PhD

Senior Editor

cschnell@plos.org

PLOS Biology

Reviewer remarks:

Reviewer #2 (Yves Boubenec): Thanks for having taken the time to address my comments in depth.

I am satisfied both by the additional analysis on the two types of electrodes, as well as you further explanations.

---

## [Editor Report · Decision Letter 4]

29 Jul 2024

Dear Dr Cleary,

Thank you for the submission of your revised Short Reports "Syllable processing is organized in discrete subregions of the human superior temporal gyrus" for publication in PLOS Biology. On behalf of my colleagues and the Academic Editor, David Poeppel, I am pleased to say that we can in principle accept your manuscript for publication, provided you address any remaining formatting and reporting issues. These will be detailed in an email you should receive within 2-3 business days from our colleagues in the journal operations team; no action is required from you until then. Please note that we will not be able to formally accept your manuscript and schedule it for publication until you have completed any requested changes.

PRESS

Sincerely, 

Christian

Christian Schnell, PhD

Senior Editor

PLOS Biology

cschnell@plos.org